# 6-Polyaminosteroid Squalamine Analogues Display Antibacterial Activity against Resistant Pathogens

**DOI:** 10.3390/ijms24108568

**Published:** 2023-05-10

**Authors:** Delphine Vergoz, Flore Nilly, Florie Desriac, Magalie Barreau, Antoine Géry, Charlie Lepetit, François Sichel, Katy Jeannot, Jean-Christophe Giard, David Garon, Sylvie Chevalier, Cécile Muller, Emmanuelle Dé, Jean Michel Brunel

**Affiliations:** 1Polymers, Biopolymers, Surfaces Laboratory, University of Rouen Normandie, INSA Rouen, CNRS, UMR 6270, 76000 Rouen, France; delphine.vergoz@univ-rouen.fr (D.V.); emmanuelle.de@univ-rouen.fr (E.D.); 2Communication Bactérienne et Stratégies Anti-Infectieuses, University of Rouen Normandie, CBSA, 27000 Evreux, France; flore.nilly@univ-rouen.fr (F.N.); magalie.barreau@univ-rouen.fr (M.B.); sylvie.chevalier@univ-rouen.fr (S.C.); 3Communication Bactérienne et Stratégies Anti-Infectieuses, UNICAEN, Normandie University, UR4312, CBSA, 14032 Caen, France; florie.desriac@unicaen.fr (F.D.);; 4UNICAEN, Normandie University, ABTE UR4651 and Centre François Baclesse, 14032 Caen, France; antoine.gery@unicaen.fr (A.G.); charlie.lepetit@unicaen.fr (C.L.); francois.sichel@unicaen.fr (F.S.); david.garon@unicaen.fr (D.G.); 5UMR 6249 Chrono-Environnement, CNRS-Université de Bourgogne/Franche-Comté, 25000 Besançon, France; katy.jeannot@univ-fcomte.fr; 6UNICAEN, University of Rouen Normandie, INSERM, DYNAMICURE UMR 1311 F, 14000 Caen, France; jean-christophe.giard@unicaen.fr; 7Aix Marseille University, INSERM, SSA, MCT, 13385 Marseille, France

**Keywords:** aminosteroid derivatives, polyamine, antimicrobial activities, *Acinetobacter baumannii*, *Pseudomonas aeruginosa*, *Staphylococcus aureus*, *Enterococcus faecium*

## Abstract

A series of 6-polyaminosteroid analogues of squalamine were synthesized with moderate to good yields and evaluated for their *in vitro* antimicrobial properties against both susceptible and resistant Gram-positive (vancomycin-resistant *Enterococcus faecium* and methicillin-resistant *Staphylococcus aureus*) and Gram-negative (carbapenem-resistant *Acinetobacter baumannii* and *Pseudomonas aeruginosa)* bacterial strains. Minimum inhibitory concentrations against Gram-positive bacteria ranged from 4 to 16 µg/mL for the most effective compounds, **4k** and **4n**, and showed an additive or synergistic effect with vancomycin or oxacillin. On the other hand, the derivative **4f**, which carries a spermine moiety like that of the natural trodusquemine molecule, was found to be the most active derivative against all the resistant Gram-negative bacteria tested, with an MIC value of 16 µg/mL. Our results suggest that 6-polyaminosteroid analogues of squalamine are interesting candidates for Gram-positive bacterial infection treatments, as well as potent adjuvants to fight Gram-negative bacterial resistance.

## 1. Introduction

The accelerated emergence of many multidrug-resistant (MDR) bacterial pathogens, due to the widespread use of antibiotics, has raised an urgent need for a renewed search for antibacterial agents [1]. Antibiotics were treated as miracle drugs when they first became available half a century ago, but their popularity rapidly led to overuse. Over the last decade, it has become obvious that antibiotics are losing their effectiveness as bacteria develop resistance against them, and new drugs rarely reach the market. Bacteria can acquire resistance to drugs in multiple ways, so circumventing the resistance problem is no small task [2,3,4,5]. Pharmaceutical companies have recently revived efforts to develop new antibiotics. However, avoiding a future in which bacteria are once again widespread killers requires more than one approach, among them: the rational use of antibiotics in health care and veterinary medicine, improved techniques for developing new drugs, and new perspectives on how to live with the infectious creatures with which we share the planet. In recent years, a wide variety of low-molecular-weight antibiotics have been isolated from various animal species [6]. Among the strategies envisioned in our laboratory for the search of new host-defense agents, squalamine **1** and trodusquemine **2** (Figure 1) were identified as the first natural aminosterols from the dogfish shark, *Squalus acanthias*, exhibiting potent antimicrobial activities against MDR bacterial strains [7,8,9].

Due to the difficulty of obtaining squalamine from natural sources, many synthetic methods have been performed using expensive starting materials [10]. For these reasons, we have envisaged designing 6-polyaminosterol squalamine mimics that are as effective as squalamine, in a small number of steps, using inexpensive commercial starting materials, and then determining their antibacterial efficacy against a wide panel of resistant Gram-positive and Gram-negative pathogens listed by the WHO and included in the ESKAPE group (*Enterococcus faecium*, *Staphylococcus aureus*, *Klebsiella pneumoniae*, *Acinetobacter baumannii*, *Pseudomonas aeruginosa* and the *Enterobacter* species). In this study, we focused on two Gram-negative (*A. baumannii* and *P. aeruginosa*) and two Gram-positive (*E. faecium* and *S. aureus*) bacteria. *E. faecium* is a gastrointestinal commensal that is generally harmless in healthy individuals, but poses a real threat to immunocompromised hosts [11]. This microorganism is intrinsically resistant to several commonly used antibiotics [12] and has a high capacity to disseminate resistance determinants [13]. The use of vancomycin as a last resort to fight enterococci not susceptible to penicillins and aminosides has led to the emergence of vancomycin-resistant enterococci (VRE). *S. aureus* is also a major human pathogen that causes chronic infections, often related to medical devices, ranging from skin or soft-tissue infections to endocarditis [14]. This microorganism has a strong ability to develop resistance to antibiotics such as gentamicin or penicillin [15]. A penicillin derivative, methicillin, was used to inhibit *S. aureus* growth by binding to penicillin-binding proteins (PBP2a), but the synthesis of a low-affinity β-lactam PBP led to methicillin-resistant *S. aureus* (MRSA). Concerning Gram-negative bacteria, *A. baumannii* and *P. aeruginosa* are the main pathogens contributing to the burden of antimicrobial resistance (AMR) [1]. Both are opportunistic pathogens responsible for hospital-associated infections, mainly hospital-acquired pneumonia, skin and soft-tissue infections or urinary tract infections [16,17]. Carbapenems are antibiotics of last resort for the treatment of these severe infections. However, the increasing number of carbapenem-resistant isolates and the associated deaths (e.g., more than 50,000 deaths attributable to carbapenem-resistant *A. baumannii* (CRAb) in 2019) are particularly problematic [1]. These pathogens develop different resistance mechanisms to carbapenems, including decreased outer membrane permeability, overproduction of efflux pumps, degrading enzymes, and/or modification of penicillin-binding proteins.

In this context, where the development of an anti-bacterial agent is a necessity, we tested the effectiveness of squalamine 6-polyaminosterol derivatives against these problematic bacteria.

## 2. Results and Discussion

### 2.1. Synthesis of New 6-Polyaminosteroid Derivatives

Using an efficient titanium reductive amination protocol developed in our laboratory [18,19], we have achieved a one-step synthesis procedure for the preparation of new polyaminosterol derivatives from commercially available 6-ketocholestanol, according to the following synthetic pathway [20].

In a preliminary attempt, we searched for the optimal experimental conditions needed to perform this reaction, using 1,2-diaminoethane as a test amine. The isolated yields of compound **4a** appear to be highly solvent dependent (Table 1).

The expected amino derivative **4a** was obtained with a 68% yield when the reaction was performed in MeOH (Table 1, entry 1), while only moderate yields of 48, 18, and 34% were achieved when the reactions were performed in CH_2_Cl_2_, toluene and THF, respectively (Table 1, entries 3–5). On the other hand, performing the reaction in MeOH and increasing the reaction temperature from −78 to 0 °C led to a significant decrease in the diastereoselectivity, from 95 to 58% de, respectively (Table 1, entries 1 and 2).

A mechanism including a nucleophilic attack of the amino group on a carbonyl compound activated by a titanium Lewis acid, involving a transient imine species which is subsequently reduced, was postulated. Rationalization of the exclusive formation of the β-polyamine derivative was achieved through a transition state model which suggests that the hydride attack occurs at the C6 carbon in an *α* position, due to steric methyl and hydrogen hindrance, generating primarily the 6β-polyamine cholestanol derivative **4** (Figure 1). Moreover, this model allowed us to explain why the formation of the 6α-parent derivative is strongly disfavored at low temperatures.

On the other hand, regardless of the nature of the diamine or polyamine considered, the expected products were obtained with chemical overall yields ranging from 28 to 98% and excellent diastereoselectivities, up to 95% in all cases (Figure 2).

### 2.2. Activities of the New 6-Polyaminosteroid Derivatives on Resistant Bacterial Strains

The eighteen 6-polyaminosteroid derivative compounds were tested for their activity against *E. faecium* and *S. aureus*, as these species can become VRE and MRSA, respectively, contributing to AMR, and belong to the MDR bacteria that sometimes lead to therapeutic impasses. Interestingly, no significant difference of Minimal Inhibitory Concentration (MIC) values (more or less than two-fold) was observed between susceptible and resistant strains for any of the molecules tested. For vancomycin-susceptible *E. faecium* (VSE) and VRE, the MICs ranged from 4 to 32 mg/L and from 2 to 16 mg/L, respectively (Table 2). Similarly, MSSA and MRSA presented MICs varying from 4 to 32 mg/L. These values appeared to be in the same order of magnitude as those measured for the princeps molecule, squalamine **1**, which were 8, 8, 4 and 2 mg/L for VSE, VRE, MSSA and MRSA, respectively (Table 2). Furthermore, the mechanisms enabling antimicrobial resistance in *E. faecium* (modification of the peptidoglycan structure due to the presence of the *van* operon) [21] and *S. aureus* (presence of PBP2a, encoded by the *mecA* gene) [22] did not interfere with the activity of the 6-polyaminosteroid derivatives tested. All these data are therefore of clinical interest in the search for new strategies to fight MDR pathogens.

As observed for the MICs, except for molecule **4n**, the MBCs of the other molecules for resistant strains (VRE and MRSA) were very similar to those of susceptible isolates (VSE and MSSA). The MBCs for enterococci (VSE and VRE) were mostly close to the MICs (equal to or two-fold higher), except for **4d**, **4k**, **4n**, **4p** and **4q**, which were 4 to 16-fold higher. The compounds derived from 6-polyaminosteroid appeared to be less bactericidal against *S. aureus* than against *E. faecium* as the MBCs were significantly higher. Molecules **4k** and **4n** were exceptions with lower or equal values.

To assess whether the new molecules could at least partially restore antibiotic susceptibility, combinations with vancomycin (for VRE) or oxacillin (for MRSA) were also tested on resistant strains (Table 2). For *Enterococcus*, six of the nineteen compounds tested (**4c**, **4e**, **4h**, **4j**, **4l** and **4m**) had no influence on vancomycin resistance or vice versa (FICI between 1 and 2). In contrast, combinations with the others, as well as with squalamine, showed an additive effect (FICI ranging from 0.51 to 0.75) (Table 2). More drastically, 17 of the 6-polyaminosteroid derivatives exhibited synergistic activity with oxacillin against *S. aureus* (FICI between 0.25 and 0.5), while the derivatives **4h**, **4q** and squalamine **1** led to an additive effect (FICI of 0.51, 0.53 and 0.75, respectively) (Table 2). The mechanism of action of squalamine against Gram-positive bacteria has been studied previously and involves strong membrane depolarization leading to ion efflux [23] while the antibiotics used interact with the cell wall. It is tempting to speculate that the additive and synergistic effects of the new molecules and the encountered antibiotics could be due to simultaneous effects against different targets, amplifying the observed response.

Of all the 6-polyaminosteroid derivatives tested, **4k** and **4n** emerged as the most promising candidates as they exhibited the lowest MICs/MBCs with VSE, VRE, MSSA and MSRA, and had additive or synergistic effects with vancomycin or oxacillin.

Several studies have shown that squalamine and its derivatives can be used as antimicrobials [24,25] or as disinfectant agents [26], but our results have revealed that squalamine-analogues could also be considered as adjuvants to antibiotic therapy.

Examination of the activity of the derivatives against Gram-negative pathogens (Table 3) shows an obvious decrease in efficacy in comparison to Gram-positive pathogens. Indeed, we can observe that a good activity is maintained against the *A. baumannii* reference strain (more than 84% of the derivatives present MICs ≤ IC_50_), but only seven derivatives (37%) remain active against the CRAb strain. Activity is even lower against *P. aeruginosa* where only three compounds (**4f**, **4l** and **4q**) succeed in stopping the growth of susceptible or resistant strains at a concentration below the IC_50_ value. In almost all cases, the MBC was identical to the MIC indicating a bactericidal mode of action for these compounds. Finally, we focused on the three compounds (**4f, 4l** and **4q**) that were both active against *A. baumannii* CRAb and *P. aeruginosa* CarbR and examined their ability to synergize with imipenem activity. Both strains exhibited imipenem MICs of 64 mg/L. As demonstrated by the calculated FICI index, only imipenem and the **4f** and **4q** derivatives presented additive activities (FICI = 0.75) when used together to inhibit bacterial growth. Of note, compound **4f**, which bears a spermine moiety like that of the natural trodusquemine molecule, was found to be more active than the squalamine analog, **4g**, which has a spermidine moiety.

As previously mentioned, the minimum inhibitory concentrations ranged from 4 to 16 mg/L for compounds **4k** and **4n**, which were the most effective against Gram-positive bacteria and showed an additive or synergistic effect with vancomycin or oxacillin. On the other hand, the derivative **4f**, which carries a spermine moiety like that of the natural trodusquemine molecule, was found to be the most active derivative, with an MIC value of 16 mg/L against all the resistant Gram-negative bacteria tested. More precisely, in this latter case, although the compounds **4f**, **4l** and **4q** contain four amino groups in their C6 side chains, **4l** and **4q** were found to exhibit significantly lower antibacterial activity than **4f** against the Gram-negative bacterial strains *A. baumannii* and *P. aeruginosa* (MICs ranging from 32–64 mg/L), suggesting that the length of the side chain and the location of the amino groups are crucial for activity, as it was also found for the derivative **4n**. The diamine derivatives showed lower activities, though still in the same ranges as all the other compounds considered. On the other hand, the added amino groups in the cyclic compound **4n** and the branched side chains of **4k** resulted in good activity, as shown by MIC values ranging from 4 to 16 mg/L for *E. faecium* and *S. aureus*. Interestingly, in the case of Gram-negative bacteria, no such side-chain-dependent effect could be observed, and the derivative **4k** represents a good compromise in terms of structure, as it has excellent to moderate activities against both Gram-positive and Gram-negative bacteria.

### 2.3. Cytotoxicity against A549 Human Lung Cells

Table 2 and Table 3 show the cytotoxicity of the compounds against the A549 human lung cell line after 24 h exposure. Squalamine has a 50% inhibitory concentration (IC_50_) of 67 mg/L, a value in good agreement with that of Carmona et al. on human pancreatic BxPC2 and hepatic Huh7 cell lines [27]. The IC_50_ of the compounds are 11.84 and 88.90 mg/L for **4i** and **4q**, respectively. Most show a cytotoxicity close to squalamine; only three show a significantly higher cytotoxicity with an IC_50_ below 17 mg/L and a NOEC (No Observed Effect Concentration) below 12.5 mg/L (**4i**, **4r**, **4p** and **4k**). Considering the most active compounds against the most susceptible bacteria (MICs ranging from 2 to 4 mg/L), all (**4d**, **4j** and **4n**) except one (**4k**) have NOECs that are clearly higher than the MICs. For the compound **4k**, only 8% growth inhibition was found at the lowest concentration tested (12.5 mg/L), suggesting no cytotoxicity at the MIC (4 mg/L). Overall, these results show good selectivity in the toxicity of these four molecules against *E. faecium* relative to A549 lung cells. In addition, the cytotoxicity against A549 cells of these four compounds was comparable to that of ciprofloxacin, as shown by Kloskowski et al. [28]. Therefore, their cytotoxicity does not seem to be a hindrance to their further development.

## 3. Material and Methods

### 3.1. Bacterial Strains and Culture Conditions

The strains used in this study are listed in Table 4. Cultures of *E. faecium, S. aureus, A. baumannii* and *P. aeruginosa* were grown in cation-adjusted Mueller-Hinton (MH) agar and broth (Biokar diagnostics/BD Difco, Sparks, MD, USA).

### 3.2. Antibacterial Assays

The Minimum Inhibitory Concentrations (MICs) of compounds were determined by the broth microdilution technique, according to the Clinical and Laboratory Standards Institute (CLSI) guidelines. For *E. faecium*, *S. aureus*, *A. baumannii* and *P. aeruginosa* inoculum, overnight cultures were diluted to an initial inoculum of 5 × 10^5^ CFU/mL. The MIC was defined as the lowest concentration that inhibited bacterial growth.

The MBC (Minimum Bactericidal Concentration) was determined directly after the MIC evaluation. Samples from each well with no bacterial growth in the microtiter plates were streaked with a calibrated inoculating loop onto MHA plates and incubated overnight at 37 °C. The MBC was determined as the concentration at which no bacterial colonies were obtained on plates.

### 3.3. Checkerboard Assays

The synergistic effect between polyaminosterol analogues and antibiotics typically used in therapy was assessed by a combination assay, as previously described [35], using oxacillin (128 to 0.25 mg/L) and vancomycin (32 to 0.06 mg/L) for MRSA and VRE, respectively, and imipenem (128 to 0.25 mg/mL) for *A. baumannii* CRAb and *P. aeruginosa* IMP. Inocula were prepared as described for MIC determination, and the highest concentration of the synthetic compound used was 2× MIC. The fractional inhibitory concentration index (FICI) was calculated as follows: FIC of the synthetic compound + FIC of the antibiotic, where FIC = MIC of the drug tested in combination/MIC of drug alone. To interpret these assays, the FICI ranges used were: ≤0.5, synergistic; between 0.5 and 1, additive; between 1 and 2, indifferent; >2 antagonistic effect.

### 3.4. Cytotoxicity Assays

A549 cells (human alveolar epithelial cells derived from an adenocarcinoma) with a doubling time of 24 h were cultured in 96-well microplates (BD Falcon) in DMEM medium (Gibco, Billings, MT, USA) supplemented with 1% bicarbonate solution at 7.5% (Gibco, Billings, MT, USA) and 10% decomplemented fetal calf serum (FCS). Each well was seeded 24 h before exposure with 10,000 cells suspended in 200 μL of medium, and then the microplates were incubated in a stove at 37 °C in a 5% CO_2_ atmosphere.

Cells were exposed for 24 h to five different dilutions of each compound (expressed in mg/L): 200, 100, 50, 25 and 12.5. Three replicates were made by dilution. After the end of the exposure time, the cells were stained with sulforhodamine B (SRB) and the absorbance reading was performed spectrophotometrically at 570 and 655 nm according to the Skehan et al. method [36]. The results were expressed as percentages of cell viability compared to the control (culture medium). The 50% inhibitory concentration (IC_50_) was defined as the concentration of the compound that induced a 50% decrease in viable cells, and the No Observed Effect Concentration (NOEC) was defined as the highest concentration inducing no growth inhibition.

## 4. Conclusions

For several years, AMR has been recognized as a global health problem, and the search for new antibacterial strategies as an urgent necessity. In this context, the search for antibacterial agents of natural origin is effective and leads to interesting results, as for squalamine [7,10]. However, there is also a need for the development of reliable and inexpensive synthetic methods to enable progress in the drug discovery process. In this study, we reported the synthesis of squalamine derivatives, 6-polyaminosterols, obtained by a fast, efficient, and inexpensive method. These 18 compounds, containing various polyamine side-chains, were tested for their antibacterial properties on some of the most problematic Gram-positive and Gram-negative resistant species to date. Most of them showed excellent activity (MIC and MBC) on *Enterococcus* and *Staphylococcus* species, regardless of the resistance phenotype. The activity on Gram-negative species is even maintained for three compounds (**4f**, **4l**, **4q**) possessing four amino groups on the C6 side chain. Several derivatives showed a synergistic effect with antibiotics currently used to fight severe infections, and as such could be very promising as adjuvants in therapeutic strategies. Finally, these results are very encouraging since the products demonstrate very good antibacterial efficiencies, comparable in many cases to squalamine [9,37] or claramine A1 [38] developed by our laboratory, while also being of major interest due to the simplicity of their synthesis, requiring one step with commercial products.

For clarity, the synthesis of all the polyaminosterol derivatives, and their relative ^1^H and ^13^C NMR spectra, were reported in the Appendix A section.

## Data Availability

Not applicable.

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
