# Peer review of "6-Polyaminosteroid Squalamine Analogues Display Antibacterial Activity against Resistant Pathogens"

_ijms, 2023, doi:10.3390/ijms24108568_

Round 1
Reviewer 1 Report
In this manuscript Dé, Brunel and cooworkers describe the synthesis of a library of (3b,6b)-6-substituted colestan-3-ol derivatives, exploiting a diastereoselective reductive amination developed in the past in their laboratories. The synthesized compounds were then tested in vitro as antibacterial agents against both Gram-positive and Gram-negative resistant bacterial strains, finding, even if not ground-breaking, interesting activities. The search for new entities able to contrast resistant bacteria is a really important task for synthetic organic chemists nowadays, antibiotic resistant infections are wide spreading causing a huge number of dead/year and there is an urgent need for treatments.
The work in the present form display issues, especially concerning the synthesis part, and could be considered for publication in IJMS after a major revision following the subsequent indications.
First of all: copies of 1H and 13C NMR spectra must be added to the Supplementary Materials for new compounds. This is mandatory.
Compound 4f, that is also the more active against Gram-negative, cannot be presented as “New” it is “Claramine” that is also commercially available.
Some of the other derivatives appears as antidiabetics in a 2013 patent (WO2013057422), however, they are not reported in any scientific article.
Then in particular:
Introduction:
- Line 48: “In recent years, a wide variety of low molecular weight antibiotics have been isolated from various animal species.” A reference should be added for this sentence.
- Line 54: “synthesizing” should be substituted with “obtaining”.
Synthesis:
- Line 113: The diastereoselectivity is ascribed to the steric hindrance of the hydrogen. But what about the methyl? It is in the same 1,3 relationship with the imine and a Me group is bulkier than an H.
- Lines 118 and 124: “Figure 2” and “Figure 3” should be “Scheme 1” and “Scheme 2”.
- Figure 2: Only a methyl is represented as “Me”. “Me” has to be added to all methyl groups or this “Me” has to be removed.
- Line 120: “yields ranging from 34 to 98%” The reported yield for 4g is 28%.
- Figure 3: diastereomeric excess % for each compound should be added in brackets next to the yield.
Activities:
- A more effective SAR discussion should be added. There are only few sentences in the comments to Table 4.
Supplementary Materials:
-Nomenclature of compounds is not correct. Names must follow IUPAC rules:
4a is “(3b,6b)-6-[(2-aminoethyl)amino]cholestan-3-ol”
4b is “(3b,6b)-6-[(3-aminopropyl)amino]cholestan-3-ol”
4e is “(3b,6b)-6-({3-[(3-aminopropyl)amino]propyl}amino)cholestan-3-ol”
4m is “(3b,6b)-6-[4-(2-aminoethyl)-1-piperazinyl]cholestan-3-ol”
And so on…
-13C NMR chemical shifts must be reported with one decimal.
-Usually, HRMS or elemental analysis are requested for new compounds, in my opinion this is not mandatory, however, images of 1H and 13C NMR spectra must be added as mentioned before.

Author Response
Dear editor,
Please find enclosed the manuscript entitled “6-polyaminosteroid squalamine analogues displaying antibacterial activities against resistant pathogens” that we have submitted to your journal International Journal of Molecular Sciences and modified according to the comments of the referees.
All the modifications are highlighted in yellow in the main text. See below the point-by-point answers to the referee’s comments.
Referee 1
The work in the present form display issues, especially concerning the synthesis part, and could be considered for publication in IJMS after a major revision following the subsequent indications.
First: copies of 1H and 13C NMR spectra must be added to the Supplementary Materials for new compounds. This is mandatory.
This part was modified by adding NMR spectra in the supporting information section and referring to our previous description of the compounds.
Compound 4f, that is also the more active against Gram-negative, cannot be presented as “New” it is “Claramine” that is also commercially available.
Changed done according to the comment of the referee
Some of the other derivatives appears as antidiabetics in a 2013 patent (WO2013057422), however, they are not reported in any scientific article.
Reference of the patent was added in the text (p.4) and in the reference section.
- Line 48: “In recent years, a wide variety of low molecular weight antibiotics have been isolated from various animal species.” A reference should be added for this sentence.
Changed done by adding reference (10.3390/molecules25102402)
- Line 54: “synthesizing” should be substituted with “obtaining”.
Changed done according to the comment of the referee
- Line 113: The diastereoselectivity is ascribed to the steric hindrance of the hydrogen. But what about the methyl? It is in the same 1,3 relationship with the imine and a Me group is bulkier than an H.
Changed done according to the comment of the referee
- Lines 118 and 124: “Figure 2” and “Figure 3” should be “Scheme 1” and “Scheme 2”.
Changed done according to the comment of the referee
- Figure 2: Only a methyl is represented as “Me”. “Me” has to be added to all methyl groups or this “Me” has to be removed.
Changed done according to the comment of the referee.
- Line 120: “yields ranging from 34 to 98%” The reported yield for 4g is 28%.
Changed done according to the comment of the referee.
- Figure 3: diastereomeric excess % for each compound should be added in brackets next to the yield.
Changed according to the comment of the referee.
- A more effective SAR discussion should be added. There are only few sentences in the comments to Table 4.
At this stage difficult to give a more effective SAR discussion
Supplementary Materials:
-Nomenclature of compounds is not correct. Names must follow IUPAC rules:
4a is “(3b,6b)-6-[(2-aminoethyl)amino]cholestan-3-ol”
4b is “(3b,6b)-6-[(3-aminopropyl)amino]cholestan-3-ol”
4e is “(3b,6b)-6-({3-[(3-aminopropyl)amino]propyl}amino)cholestan-3-ol”
4m is “(3b,6b)-6-[4-(2-aminoethyl)-1-piperazinyl]cholestan-3-ol”
And so on…
The nomenclature of the compounds was changed according to the recommendation of the referee.
-Usually, HRMS or elemental analysis are requested for new compounds, in my opinion this is not mandatory, however, images of 1H and 13C NMR spectra must be added as mentioned before.
This part was modified by adding NMR spectra in the supporting information section and referring to our previous description of the compounds.
Referee 2
The authors reported the synthesis of 6-polyaminosteroid squalamine as an antimicrobial agent. The submission can be accepted after revision considering the following points:
The title should be revised to be short, precise, and informative. Redundant words such as ‘New’, and ‘against both Gram-positive and Gram-negative resistant bacteria’ should be removed.
The title of the manuscript was changed according to the recommendation of the referee.
- The characterization techniques such as 1H NMR, 13C NMR should be added to the main text confirming the synthesis of the proposed molecules.
The addition of the NMR spectra in the supporting information section was reported according to the comment of the referee.
- The mechanism of antimicrobial action should be discussed and supported with experimental data.
The understanding of the mechanism of action of these compounds (compared to squalamine) is underway using different techniques on different bacterial strains. These results will be the subject of a separate publication in preparation.
- The biodegradation of the molecules should be investigated.
Not studied for the moment but this has been already done for squalamine (Non-Genotoxic Assessment of a Natural Antimicrobial Agent: Squalamine Anti-infective agents 2014, 12, 75) demonstrating a great stability in various experimental conditions. We suspect that we will observe quite similar results for these compounds (to be reported elsewhere).
- A comparison with previously published materials should be discussed and summarized in a Table. A sentence was added in the conclusion for comparison with other published materials
- References for antibacterial materials should be updated suggesting these examples; International Journal of Molecular Sciences 2022, 23 (10), 5405; Int. J. Mol. Sci.2022, 23(1), 545; Frontiers in Chemical Engineering 2021, 3, 790314; Int. J. Mol. Sci. 2022, 23(6), 3209
Do not correspond to our approach
- The language should be revised and typos should be corrected.
Typos and grammar were carefully checked.
Hoping that this manuscript will be now accepted for publication in International Journal of Molecular Sciences.Thank you for your and the reviewer’s consideration. Very truly yours. JM. Brunel
Reviewer 2 Report
The authors reported the synthesis of 6-polyaminosteroid squalamine as an antimicrobial agent. The submission can be accepted after revision taking into account the following points:-
1. The title should be revised to be short, precise, and informative. Redundant words such as ‘New’, and ‘against both Gram-positive and Gram-negative resistant bacteria’ should be removed.
2. The characterization techniques such as 1H NMR, 13 CNMR should be added to the main text confirming the synthesis of the proposed molecules.
3. The mechanism of antimicrobial action should be discussed and supported with experimental data.
4. The biodegradation of the molecules should be investigated.
5. A comparison with previously published materials should be discussed and summarized in a Table.
6. References for antibacterial materials should be updated suggesting these examples; International Journal of Molecular Sciences 2022, 23 (10), 5405; Int. J. Mol. Sci. 2022, 23(1), 545; Frontiers in Chemical Engineering 2021, 3, 790314; Int. J. Mol. Sci. 2022, 23(6), 3209
7. The language should be revised and typos should be corrected.
The authors reported the synthesis of 6-polyaminosteroid squalamine as an antimicrobial agent. The submission can be accepted after revision taking into account the following points:-
1. The title should be revised to be short, precise, and informative. Redundant words such as ‘New’, and ‘against both Gram-positive and Gram-negative resistant bacteria’ should be removed.
2. The characterization techniques such as 1H NMR, 13 CNMR should be added to the main text confirming the synthesis of the proposed molecules.
3. The mechanism of antimicrobial action should be discussed and supported with experimental data.
4. The biodegradation of the molecules should be investigated.
5. A comparison with previously published materials should be discussed and summarized in a Table.
6. References for antibacterial materials should be updated suggesting these examples; International Journal of Molecular Sciences 2022, 23 (10), 5405; Int. J. Mol. Sci. 2022, 23(1), 545; Frontiers in Chemical Engineering 2021, 3, 790314; Int. J. Mol. Sci. 2022, 23(6), 3209
7. The language should be revised and typos should be corrected.
Author Response
Dear editor,
Please find enclosed the manuscript entitled “6-polyaminosteroid squalamine analogues displaying antibacterial activities against resistant pathogens” that we have submitted to your journal International Journal of Molecular Sciences and modified according to the comments of the referees.
All the modifications are highlighted in yellow in the main text. See below the point-by-point answers to the referee’s comments.
Referee 1
The work in the present form display issues, especially concerning the synthesis part, and could be considered for publication in IJMS after a major revision following the subsequent indications.
First: copies of 1H and 13C NMR spectra must be added to the Supplementary Materials for new compounds. This is mandatory.
This part was modified by adding NMR spectra in the supporting information section and referring to our previous description of the compounds.
Compound 4f, that is also the more active against Gram-negative, cannot be presented as “New” it is “Claramine” that is also commercially available.
Changed done according to the comment of the referee
Some of the other derivatives appears as antidiabetics in a 2013 patent (WO2013057422), however, they are not reported in any scientific article.
Reference of the patent was added in the text (p.4) and in the reference section.
- Line 48: “In recent years, a wide variety of low molecular weight antibiotics have been isolated from various animal species.” A reference should be added for this sentence.
Changed done by adding reference (10.3390/molecules25102402)
- Line 54: “synthesizing” should be substituted with “obtaining”.
Changed done according to the comment of the referee
- Line 113: The diastereoselectivity is ascribed to the steric hindrance of the hydrogen. But what about the methyl? It is in the same 1,3 relationship with the imine and a Me group is bulkier than an H.
Changed done according to the comment of the referee
- Lines 118 and 124: “Figure 2” and “Figure 3” should be “Scheme 1” and “Scheme 2”.
Changed done according to the comment of the referee
- Figure 2: Only a methyl is represented as “Me”. “Me” has to be added to all methyl groups or this “Me” has to be removed.
Changed done according to the comment of the referee.
- Line 120: “yields ranging from 34 to 98%” The reported yield for 4g is 28%.
Changed done according to the comment of the referee.
- Figure 3: diastereomeric excess % for each compound should be added in brackets next to the yield.
Changed according to the comment of the referee.
- A more effective SAR discussion should be added. There are only few sentences in the comments to Table 4.
At this stage difficult to give a more effective SAR discussion
Supplementary Materials:
-Nomenclature of compounds is not correct. Names must follow IUPAC rules:
4a is “(3b,6b)-6-[(2-aminoethyl)amino]cholestan-3-ol”
4b is “(3b,6b)-6-[(3-aminopropyl)amino]cholestan-3-ol”
4e is “(3b,6b)-6-({3-[(3-aminopropyl)amino]propyl}amino)cholestan-3-ol”
4m is “(3b,6b)-6-[4-(2-aminoethyl)-1-piperazinyl]cholestan-3-ol”
And so on…
The nomenclature of the compounds was changed according to the recommendation of the referee.
-Usually, HRMS or elemental analysis are requested for new compounds, in my opinion this is not mandatory, however, images of 1H and 13C NMR spectra must be added as mentioned before.
This part was modified by adding NMR spectra in the supporting information section and referring to our previous description of the compounds.
Referee 2
The authors reported the synthesis of 6-polyaminosteroid squalamine as an antimicrobial agent. The submission can be accepted after revision considering the following points:
The title should be revised to be short, precise, and informative. Redundant words such as ‘New’, and ‘against both Gram-positive and Gram-negative resistant bacteria’ should be removed.
The title of the manuscript was changed according to the recommendation of the referee.
- The characterization techniques such as 1H NMR, 13C NMR should be added to the main text confirming the synthesis of the proposed molecules.
The addition of the NMR spectra in the supporting information section was reported according to the comment of the referee.
- The mechanism of antimicrobial action should be discussed and supported with experimental data.
The understanding of the mechanism of action of these compounds (compared to squalamine) is underway using different techniques on different bacterial strains. These results will be the subject of a separate publication in preparation.
- The biodegradation of the molecules should be investigated.
Not studied for the moment but this has been already done for squalamine (Non-Genotoxic Assessment of a Natural Antimicrobial Agent: Squalamine Anti-infective agents 2014, 12, 75) demonstrating a great stability in various experimental conditions. We suspect that we will observe quite similar results for these compounds (to be reported elsewhere).
- A comparison with previously published materials should be discussed and summarized in a Table. A sentence was added in the conclusion for comparison with other published materials
- References for antibacterial materials should be updated suggesting these examples; International Journal of Molecular Sciences 2022, 23 (10), 5405; Int. J. Mol. Sci.2022, 23(1), 545; Frontiers in Chemical Engineering 2021, 3, 790314; Int. J. Mol. Sci. 2022, 23(6), 3209
Do not correspond to our approach
- The language should be revised and typos should be corrected.
Typos and grammar were carefully checked.
Hoping that this manuscript will be now accepted for publication in International Journal of Molecular Sciences.Thank you for your and the reviewer’s consideration. Very truly yours. J. M. Brunel
Round 2
Reviewer 1 Report
With the changes done I can now recommend this manuscript for publication in IJMS.